# Linking Perceived Policy Effectiveness and Proenvironmental Behavior: The Influence of Attitude, Implementation Intention, and Knowledge

**DOI:** 10.3390/ijerph18062910

**Published:** 2021-03-12

**Authors:** Huilin Wang, Jiaxuan Li, Aweewan Mangmeechai, Jiafu Su

**Affiliations:** 1International College, National Institute of Development Administration, 118 Moo3, Sereethai Road, Klong-Chan, Bangkapi, Bangkok 10240, Thailand; huilin.wan@stu.nida.ac.th; 2School of Design, East China Normal University, No.3663, Zhongshan Road (N), Shanghai 200062, China; lijialeixuan0306@foxmail.com; 3National Research Base of Intelligent Manufacturing Service, Chongqing Technology and Business University, Chongqing 400067, China

**Keywords:** perceived policy effectiveness, the theory of planned behavior, waste sorting and management policy, proenvironmental behavior

## Abstract

Residents’ behavior is the result of the combined effect of external environment factors and internal psychological factors. Based on the theory of planned behavior (TPB) and the attitude–behavior–condition (ABC) theory, this study aims to explore the impact of policy support on residents’ psychological factors and proenvironmental behavior. This study developed an extended TPB and ABC model and replaced the behavioral intention in the TPB model with implementation intentions to enhance the ability of the variables to explain and predict proenvironmental behavior. The longitudinal research method was adopted to collect data through a two-stage questionnaire survey of 1145 Shanghai residents. Results demonstrated that perceived policy effectiveness has a significant and positive impact on attitude, implementation intention, and proenvironmental behavior. This means that proenvironmental behavior tends to appear in people with a high perception of policy effectiveness, positive attitude, and strong implementation intention. Moreover, this study points out for the first time that high waste management knowledge weakens the relationship between perceived policy effectiveness and attitude. For residents with high waste management knowledge, the effect of simple policy publicity is limited. The findings suggest that the government should increase the breadth and depth of policy support and policy publicity to cover the entire waste management process.

## 1. Introduction

Human activities are the main factor in accelerating environmental pollution and destruction [1]. Due to rapid population growth, urbanization, and the improvement of people’s quality of life, the amount of municipal solid waste (MSW) has also increased dramatically. Shanghai’s MSW production ranked first among China’s 200 large and medium-sized cities in 2018, at 9.294 million tons [2]. Waste production will continue to maintain an annual growth rate of 8–10% [3]. The impact of waste on the environment depends a great deal on how it is processed. Currently, China’s waste disposal is dominated by landfills and incineration [4]. However, there are some problems with these two methods, of which neither represent an ideal waste disposal mode. In one respect, the leachate produced by landfill causes corresponding pollution problems by contact with the surrounding soil, ground, or surface water [5]. Additionally, untreated or improperly disposed contaminants in landfills or incineration plants threaten human health and increase the risk of cancer in nearby residents [6].

Faced with the contradiction that waste disposal capacity cannot keep up with ever-increasing waste output, increasing the government’s investment in terminal disposal facilities can only solve part of the problem. To reduce the difficulty of waste disposal and increase the service life of waste disposal facilities, it is an inevitable choice for the government to advocate source separation. From 1 July 2019, the Shanghai Municipal Solid Waste Management Regulation was officially implemented as the most stringent waste-sorting measure in Chinese history, which indicates that the era of widespread mandatory waste sorting in China has arrived [7]. As a result, Shanghai became the first city in China to comprehensively implement waste sorting and management policies, and the enforcement of the policies was strong and effective. Compared to many Chinese cities, Shanghai can undoubtedly be regarded as China’s leading city in waste management. Meanwhile, Shanghai’s experience in waste management is gradually spreading to other Chinese cities, and other developing countries should find Shanghai’s experience worth learning from.

Studies on environmental issues have received much attention in recent decades [8]. However, people’s proenvironmental behavior is not necessarily accompanied by a change in environmental concern [9]. Part of the reason is that when individuals choose whether to take proenvironmental action, they often fall into a conflict between direct personal interests and long-term collective interests [10]. Similarly, proenvironmental behavior does not directly benefit individuals but benefits others or the environment [11]. Many scholars recognize this fact and focus their research on social and psychological factors that influence personal environmental attitudes and behavior [12]. Therefore, the objectives of this study are as follows: (1) To understand the waste-sorting behavior of Shanghai residents under the guidance of the mandatory waste sorting policy; (2) to explore the factors that influence the proenvironmental behavior of Shanghai residents; (3) to inform the Shanghai government on existing problems and make suggestions.

Environmental knowledge can only explain a small part of proenvironmental behaviors, and 80% of the factors that contribute to proenvironmental behavior seem to be situational factors and other internal factors [13]. Different from previous studies that use knowledge as the antecedent of behavior, knowledge is only used as a moderating variable to influence the extended model of the theories of planned behavior (TPB) and attitude–behavior–condition (ABC) in this study. Moreover, to enhance the ability to explain and predict proenvironmental behavior, this study replaced behavioral intention in the TPB model with implementation intention. Previous studies have suggested that the relationship between external factors and psychological factors is ambiguous [14]. However, this study found the influence of extrinsic motivation (i.e., perceived policy effectiveness) on intrinsic motivation (i.e., attitude) based on the ABC theory and further stimulated the occurrence of implementation intention and proenvironmental behavior. Perceived policy effectiveness can even skip the psychological factor of attitude, which directly affects implementation intention and proenvironmental behavior. The proponents chose Shanghai residents because Shanghai has relatively extensive waste management experience and was the first Chinese city to introduce a mandatory waste sorting policy at the legislative level. Understanding the proenvironmental behavior of residents will help the development of related theories and help policymakers formulate related interventions.

The following sections are structured as follows: Section 2 reviews the literature related to TPB, ABC theory, and Shanghai waste-sorting policy instruments. Section 3 presents hypotheses and conceptual models. Section 4 introduces data collection methods, questionnaire composition, and data analysis methods. Section 5 describes the results of the data analysis and tests the hypotheses. Section 6 discusses the results and proposes practical implications. Section 7 summarizes the central ideas of the paper and proposes prospects for future research.

## 2. Literature Review

### 2.1. Theory of Planned Behavior

In the theory of planned behavior (TPB), attitude, subjective norm, and perceived behavioral control are the three main variables that determine behavioral intentions. The more positive the attitude, the greater the support of important people, the stronger the perceived behavioral control, and the stronger the behavioral intention [15]. Attitude, subjective norm, and perceived behavioral control explained 27% of the variance in behavior and 39% of the variance in behavioral intention [16]. Although TPB can explain part of behavior variance, most behavior variance still cannot be explained. In recent years, research has been devoted to improving the explanatory power of the TPB model, such as exploring new moderators and independent variables and adding implementation intention as a mediator to bridge the intention–behavior gap [17].

The difference between implementation intention and behavioral intention is that individuals with implementation intention usually have a clear plan for when, where, and how to perform their actions [18]. Implementation intention usually specifies self-relevant behavior prompts, not only clarifying when and where the behavior occurs but also adopting an if(situation)–then(behavior) format [19]. For example, in this study, the statement about implementation intention is, “For the next garbage discard, I plan to put wet waste into the designated trash can within the stipulated time”. This statement not only emphasizes the context but also explains the details. Compared with general behavioral intentions (e.g., I want to participate in waste sorting), implementation intentions can increase the achievement rate of goals [20]. Therefore, this study chose implementation intention instead of behavioral intention to enhance the ability to explain and predict proenvironmental behavior.

### 2.2. Attitude–Behavior–Condition Theory

TPB pays more attention to the influence of psychological factors on behavior but ignores the promotion or inhibition of external conditions. The limitation leads to an insufficient understanding of the determinants of proenvironmental behavior. Guagnano, Stern, and Dietz [21] put forward the attitude–behavior–condition (ABC) theory when studying the behavior of household waste recycling and believed that proenvironmental behavior is the result of the interaction of individual attitudes and external conditions. According to ABC theory, factors affecting behavior include attitude factors (e.g., norms, beliefs, values), situational factors (e.g., interpersonal influences, social expectations, advertising, government regulations, economic incentives, and costs), personal abilities (e.g., knowledge, specific actions, time allowance, money, social status, social–demographic variables), and habits or procedures [22]. A behavior occurs when the cumulative effect of external conditions and attitudes is positive [23]. Ertz, Karakas, and Sarigöllü [24] proposed that ABC theory’s attempt at perceived situational factors is a valuable exploration because it suggests that researchers examine subjectively perceived situational factors. Empirically, ABC theory has been applied in recent research on waste separation and recycling [23], consumer purchase behavior [25], and energy consumption [26]. Therefore, based on the ABC theory, this study added perceived policy effectiveness as an external environmental factor and knowledge as a personal ability factor.

### 2.3. Waste-Sorting Policy Instruments in Shanghai

Command and control instruments are widely used in China’s waste management policy and play a leading role. The advantage of these instruments is that the government can mobilize resources by implementing laws, regulations, standards, and rules and can respond directly and quickly to environmental issues [27]. In 2017, “Implementing the Classification System for Municipal Solid Waste”, a command-and-control policy, was enforced in 46 major cities and set overall targets for waste management [28]. Afterward, cities across the country began to gradually implement waste sorting under the direction of the State Council. On 1 July 2019, the Shanghai Municipal Solid Waste Management Regulation came into effect, marking the first local mandatory regulation for waste sorting in China, indicating that waste sorting has been incorporated into the Shanghai legal framework [29]. Most of China’s policies are pushed from the central government to local governments in this way. Command-and-control instruments, on the one hand, can motivate local Chinese officials to strictly enforce environmental tasks; however, excessive reliance on binding environmental objectives as China’s main policy instruments will have many negative effects [30]. Command-and-control policies are rigorous, but they do not fully address the environmental problems that are faced today [31]. 

Economic incentives mean that environmental and economic policies can be tightly integrated; these are often used in conjunction with regulations and other policy instruments. Economic incentive instruments (e.g., paying for what you throw away) are often used to encourage household waste reduction and recycling in developed countries [32]. Through the operation of the invisible hand of the market, rational and selfish individuals are driven by economic incentives to reduce waste generation or increase recycling [33]. Shanghai designed the “Green Account” to encourage residents to sort and recycle waste so that they can store points in their accounts, which can be used in exchange for physical objects (e.g., detergents, garbage bags). Correspondingly, there are certain penalties for residents and organizations that do not cooperate with this policy. Individuals face a fine of 50–200 CNY (7–30 USD) for not sorting waste properly [34].

Public engagement instruments are different from mandatory regulations and the driving force of economic interests. Their benefits include increasing public trust and the understanding of government policies to achieve more satisfying and easier decisions [35]. Shanghai has been openly recruiting volunteer teams for waste sorting and accepting applications from citizens. Volunteers take turns on duty in each community, instructing and supervising residents on how to sort waste. The number of volunteers is expected to exceed one million by 2021. For waste management policies to be effective in the implementation process, multiple policy instruments need to work together.

## 3. Hypotheses

### 3.1. Perceived Policy Effectiveness, Implementation Intention, and Proenvironmental Behavior

Policies and regulations can guide or restrict behavior to a certain extent [36]. Public authorities often shape people’s behavior through policy instruments, including mandatory rules, rewards, punishment, education, and facility building [37]. For instance, proenvironmental policy instruments are often used to punish actors for their nonenvironmentally friendly behavior [31]. Economic incentives can directly encourage people to choose proenvironmental behavior without having to go through the impact of environmental concerns [13]. Even if laws or regulations are in force, waste-sorting practices may not meet the goals set by the government because residents’ perceptions of policies vary from person to person. Thus, perceived policy effectiveness is defined by Wan, Shen, and Yu [38] as an individual’s evaluation of the clarity, adequacy, and facilitation of the policy. The more people believe that public policies are achieving their stated goals, the more likely they are to have strong intentions and take action [39]. Wan, Shen, and Yu [40] found that perceived policy effectiveness explained 19.39% of recycling intention in a model with 198 Hong Kong residents as a sample. Furthermore, Wang and Mangmeechai [17] collected survey feedback from 3113 residents of Changsha, China, and verified that perceived policy effectiveness has a significant and positive impact on implementation intention and proenvironmental behavior. Therefore, this study proposes Hypothesis 1 and Hypothesis 2.

**Hypotheses** **1** **(H1).***Perceived policy effectiveness has a positive impact on implementation intention*.

**Hypotheses** **2** **(H2).***Perceived policy effectiveness has a positive impact on proenvironmental behavior*.

### 3.2. The Mediating Roles of Attitude and Implementation Intention

ABC theory suggests modeling behavior as a multifaceted interactive result of intrinsic factors and external environmental factors [41]. Attitude was regarded as a mediator of behavioral factors in the extended ABC model in Ngah et al.’s [42] research. Based on the survey data of 709 residents in Suzhou, China, Meng et al. [43] pointed out that residents’ waste classification and recycling behaviors were significantly related to intrinsic factors (e.g., attitude) and external environmental factors (e.g., environmental knowledge, environmental facilities and services), but the combined effect of the latter was almost twice that of the former. Perceived policy effectiveness and attitude can be considered the extrinsic and intrinsic factors of behavior, respectively. Accordingly, when people feel that policy support (i.e., extrinsic motivational factors) is perfect or attractive [38], their attitude (i.e., intrinsic motivational factors) towards waste sorting will be positive. Kollmuss and Agyeman [13] believe that people’s attitudes towards environmental protection may depend on their understanding of environmental issues. When people obtain environmental information through effective channels, this may help them strengthen their control and form a positive environmental attitude. Shipeng et al.’s [44] research also found that external conditions can directly or indirectly affect psychological cognition (e.g., attitude, perceived behavioral control). The impact of policy support on implementation intentions may, therefore, be partially mediated by attitudes, for example, when residents realize that if they properly separate and place waste, following the policy, they can get points from the “Green Account” that can be exchanged for daily necessities. They may think that sorting waste is worth doing, even if it is a time-consuming task. Moreover, intention plays a mediating role in the relationship between attitude and behavior in TPB [15]. Okumah and Ankomah-Hackman’s [22] research does not support the relationship between proenvironmental attitudes and behavior because the relationship between them may be mediated by intention. Coupled with the impact of perceived policy effectiveness on attitudes, it can be connected to an external-internal behavior mechanism. This paper thus proposes the following two hypotheses: 

**Hypotheses** **3** **(H3).***Attitude mediates the relationship between perceived policy effectiveness and implementation intention*.

**Hypotheses** **4** **(H4).***Attitude and implementation intention mediate the relationship between perceived policy effectiveness and proenvironmental behavior*.

### 3.3. The Moderating Role of Knowledge

Although a high degree of perception of policy effectiveness provides support for proenvironmental behavior, residents with different levels of waste-sorting knowledge may perform differently in proenvironmental behavior. Residents with more knowledge of waste management may have a more positive attitude towards waste sorting [45] because they have actively or passively learned knowledge related to waste management (e.g., methods, importance, benefit). Okumah and Ankomah-Hackman’s [22] research also proved that the knowledge of water pollution sources affects people’s attitudes towards water resource management. According to TPB, if residents have a more positive attitude towards waste sorting, then their intention to participate in waste-sorting activities will be stronger. However, it is worth noting that not all people with positive attitudes have an intention to participate in waste sorting because people’s intentions may be affected by practical factors such as time, budget, knowledge, and available power [24,46]. Assuming that people have a positive attitude towards waste sorting but do not have relevant knowledge of waste management and do not know how to sort waste, then their willingness to participate in waste sorting may be very low because knowledge and attitude are two important factors that determine individual behavior [47]. Additionally, if residents have a wealth of waste-sorting knowledge, the impact of implementation intention on proenvironmental behavior will increase because the residents with high knowledge will encounter less resistance in their actions than those with lower knowledge. Research by Hadrich and Van Winkle [48] found that people’s understanding of environmental issues and action strategies not only cultivates people’s positive environmental attitudes but also has an impact on people’s measures to improve water quality. This study thus proposes the following hypotheses:

**Hypotheses** **5** **(H5).***The positive relationship between perceived policy effectiveness and attitude is stronger among residents with high knowledge versus those with low knowledge*.

**Hypotheses** **6** **(H6).***The positive relationship between attitude and implementation intention is stronger among residents with high knowledge versus those with low knowledge*.

**Hypotheses** **7** **(H7).***The positive relationship between implementation intention and proenvironmental behavior is stronger among residents with high knowledge versus those with low knowledge*.

A summary of all hypotheses is shown in Figure 1.

## 4. Methods

### 4.1. Participants and Procedure

This study adopted the cluster sampling method, taking the group as the sampling unit to draw samples, and each group is required to have good representativeness. The researchers selected residents of 5 districts from Shanghai’s 16 districts, based on their geographical distribution, as the research objects (see Table 1). With the help of communities, enterprises, government agencies, and schools in these five municipal districts, the researchers conducted a questionnaire survey from July to August 2020. The questionnaires were distributed to the respondents through two-stage surveys. The reason for collecting data twice, instead of collecting data only once, was to reduce the possibility of common method bias [49]. In the first stage, the respondents were required to complete surveys related to perceived policy effectiveness, attitudes, knowledge, and implementation intention. One week later, the respondents were invited again to participate in the second stage of the survey, giving feedback on their behavior in the previous week. After the two surveys, the respondents selected one item of daily necessity (e.g., waste collection bags, soap, washing-up gloves) as a reward for their participation. A total of 2000 questionnaires were distributed, and 1894 valid questionnaires were returned in the first stage, with a response rate of 94.70%. Of the 1894 questionnaires distributed in the second stage, 1145 valid questionnaires were returned; the response rate was 60.45%. Therefore, the sample loss rate from the first stage to the second stage was 39.55%. 

Table 2 lists the demographic characteristics of the residents who participated in the survey. Among the respondents, (1) 33.0% of respondents surveyed were aged 29–44 years, and 27.2% of respondents were aged 18–28 years; (2) on gender, the ratio of male and female were roughly equal (male = 52.6%, female = 47.4%); (3) on education level, the majority of respondents surveyed had a college/university degree or above (80.2%); (4) on income level, more than half of the respondents had a salary of 5000–20,000 CNY (715–2860 USD), and 35.2% of respondents earned less than 5000 CNY (715 USD) per month. The results of this survey were pretty close to the population statistics of Shanghai in 2019 [50]. However, more than 80% of the respondents reported that they had a college/university degree or above, which means that the education level reported by the sample may be higher than in the actual statistics.

### 4.2. Measures

To measure perceived policy effectiveness, seven items were extracted from the research of Wan et al. [38] and Wan et al. [40] (see Table 3). The measurement of residents’ attitudes towards waste separation and recycling was derived from Tonglet, Phillips, and Read [51]. Implementation intention was measured by using the three items of the scale derived from Gollwitzer and Brandstätter [52]. Among them, the statements of IMP1, IMP2, and IMP3 are designed based on “When I encounter the situational context Y, I will perform behavior Z” [52] (p. 188) and specific research situations. Knowledge was measured using the 3-item scale developed by Tonglet et al. [51]. Sample items include “I know what items of household waste can be recycled”, “I know how to sort household waste”, and “I know where to take household waste for recycling”. The items in the above four constructs were all measured using a five-point Likert scale, in which the responses ranged from 1 (i.e., strongly disagree) to 5 (i.e., strongly agree). Proenvironmental behavior was measured using the scale developed by Cleveland, Kalamas, and Laroche [53]. These five items utilized a five-point Likert scale from 1 (i.e., never) to 5 (i.e., very frequently).

To adapt to the research field and specific cultural background, the researchers made certain adjustments to the items of the scales. A pilot test was used to ensure the reliability of the adjusted test scale [54]. The pilot test was conducted on urban residents as the survey subjects. The researchers distributed 60 questionnaires using convenient sampling methods; 53 valid questionnaires were returned. The results showed that Cronbach’s α coefficients were all higher than 0.9, indicating that the measurement instruments have excellent internal consistency [55].

### 4.3. Data Analysis

This study used structural equation modeling (SEM) with AMOS 23.0 (via maximum likelihood estimation) to analyze the proposed model. SEM is often used to evaluate latent variables on measurement models and test hypotheses between latent variables on structural models [56]. This study adopted the two-step modeling method (i.e., using SEM to evaluate the measurement model and the structural model) suggested by Anderson and Gerbing [57]. First, we evaluated the validity of the model; then, we measured the fitting coefficient and path coefficient of the hypothetical model. Moreover, the SPSS PROCESS macro was used to test the moderated mediation model.

## 5. Results

### 5.1. Proenvironmental Behavior

Table 3 shows the proenvironmental behaviors of residents in the past week, as reported by Shanghai residents. More than 90% of Shanghai residents often or always sort wet waste and dry waste (see Table 4), which is much higher than 50.0% of Changsha residents [17] and 53.5% of Xiamen residents [58]. More than 80% of Shanghai residents claimed that they often or always sort recyclables and separate waste for recycling purposes. Based on the above data, it can be seen that most Shanghai residents sorted wet waste and dry waste because these two types of waste are the mandatory key management objects for policy implementation. In contrast, recyclable waste showed the lowest percentage of waste sorting. Thus, sorting recyclable waste should be considered the weakest aspect of Shanghai’s waste management.

### 5.2. Measurement Model

Fornell and Larcker [55] suggested that the reliability analysis should include measuring the Cronbach’s α coefficient and the composite reliability (CR) coefficient of the latent variables. The reliability test, summarized in Table 3, shows that the Cronbach’s α coefficients of the variables were in the range of 0.889–0.920, which was much higher than the recommended value of 0.7. The CR coefficients of the variables were in the range of 0.890–0.925, which was much higher than the value of 0.7, recommended by Joseph F et al. [59]. Therefore, the reliability of all variables was good. Convergent validity refers to the degree of similarity of measurement results when different measurement methods are used to determine the same feature. It is usually measured by factor loading and average variance extracted (AVE) [55]. The results in Table 3 show that the factor loading of all measurement items was between 0.574–0.894, and the AVE of all variables was between 0.628–0.762, which was higher than the recommended value of 0.5, suggested by Fornell and Larcker [55]. Therefore, all variables have high convergent validity. Additionally, researchers usually verify the discriminant validity of the data by comparing the correlation coefficient of each variable with the square root of the AVE. The results are presented in Table 5; all correlation coefficients were less than the square root of the AVE. Therefore, the variables have good discriminant validity.

### 5.3. Structural Path Model

Since neither the error term nor the residual term of the structural model has negative values, it shows that the whole model does not violate the basic fitness test criterion. With reference to the suggested value of Joseph F Hair et al. [59], the structural model showed a good fit with the data (χ^2^/df = 3.740, GFI = 0.955, NFI = 0.970, CFI = 0.978, TLI = 0.974, RMSEA = 0.049). Table 5 lists the mean, standard deviation, and correlation among the variables. Significant and positive correlations were found between the independent variables, the mediators, and the dependent variables, which provides preliminary support for the verification of the research hypotheses. The structural path model results are presented in Figure 2; the effect of perceived policy effectiveness on proenvironmental behavior was statistically significant (*β* = 0.128, *p* < 0.001), supporting H1; the effect of perceived policy effectiveness on implementation intention was statistically significant (*β* = 0.541, *p* < 0.001), supporting H2.

The conceptual model suggests that perceived policy effectiveness has a positive impact on proenvironmental behavior through two mediators (i.e., attitude and implementation intention). This study followed the recommendations of Bollen and Stine [60] and used the bootstrapping approach to verify the mediating effects. The results of 5000 bootstrap samples, with a 95% confidence interval, are presented in Table 6; all Z values were greater than 1.96, and there was no zero value in the 95% confidence interval. Moreover, it showed that significant mediation occurred between perceived policy effectiveness and implementation intention through attitude (standardized indirect effect = 0.178, *p* < 0.001), which provides support to H3. It also showed that significant mediation occurred between perceived policy effectiveness and proenvironmental behavior through attitude and implementation intention (standardized indirect effect = 0.454, *p* < 0.001), which provides support to H4. The findings mean that people who have a high perception of the effectiveness of the policy, a positive attitude, and a strong implementation intention are more likely to engage in environmental protection behaviors.

To test the moderating effect, this study followed the suggestions of Evans [61] and used hierarchical regression. First, all variables were standardized to reduce the potential impact of multicollinearity [62]. Next, the independent variable and the moderator were used as a block in Step 1, and then the interaction was used as input in Step 2 (see Table 7). The interaction term between perceived policy effectiveness and knowledge was a significant and negative predictor of attitude (*β* = −0.082, *p* < 0.000); thus, Hypothesis 5 is rejected. The interaction term between attitude and knowledge was an insignificant and positive predictor of implementation intention (*β* = 0.001, *p* > 0.05); thus, Hypothesis 6 is rejected. The interaction term between implementation intention and knowledge was a significant and positive predictor of proenvironmental behavior (*β* = 0.033, *p* < 0.05); thus, Hypothesis 7 is supported. Moreover, commonly used diagnostic methods for multicollinearity include checking tolerance and the variance inflation factor (VIF). The tolerance values of all variables were higher than 0.1, and the VIF values of all variables were lower than the maximum threshold of 5, as suggested by Rogerson [63]. Therefore, multicollinearity is not an issue in this study.

According to Preacher, Curran, and Bauer’s [64] suggestion, this study used the SPSS PROCESS macro, and slope analysis was used for cross-level interactions to test whether the significant interaction effects were consistent with the hypothetical model. As shown in Figure 3, the slope of high knowledge (M + 1SD) is gentle, while the slope of low knowledge (M − 1SD) is steep, which shows that high knowledge has a weakening effect compared to low knowledge. Therefore, the slope for residents with a low knowledge was positive and significant (*β* = 0.4060, SE = 0.0315, t = 12.8835, *p* < 0.000), whereas the slope for residents with high knowledge was weak and significant (*β* = 0.2423, SE = 0.0366, t = 6.6135, *p* < 0.000). Therefore, the above findings reject H5.

As shown in Figure 4, the slope of low knowledge (M − 1SD) is gentle, while the slope of high knowledge (M + 1SD) is steep, which shows that the low knowledge has a weakening effect compared to high knowledge. Therefore, the slope for residents with a high knowledge was positive and significant (*β* = 0.5886, SE = 0.0349, t = 16.8547, *p* < 0.000), whereas the slope for residents with a low knowledge was weak and significant (*β* = 0.5234, SE = 0.0269, t = 19.4627, *p* < 0.000). Therefore, the above findings support H7.

## 6. Discussion

### 6.1. Contributions

This study has made the following contributions to research on proenvironmental behavior. First, the researchers developed an extended TPB and ABC model to discuss the influence of external factors on psychological factors and behavior. The results indicate that perceived policy effectiveness has a significant and positive impact on attitude, implementation intention, and proenvironmental behavior. The results are consistent with previous studies [17,40]. Perceived policy effectiveness has the strongest impact on attitude and implementation intention, while its impact on proenvironmental behavior is relatively weak because the impact of perceived policy effectiveness on proenvironmental behavior is mediated by attitude and implementation intention. The findings support the contents of TPB and ABC theories [42]. As presented in Figure 2, variables can explain 53% of the variance in proenvironmental behavior, which is much higher than the 20–30% in previous studies [65]. This shows that the relationship between external factors and psychological factors is not vague. External factors (i.e., perceived policy effectiveness) not only affect psychological factors (i.e., attitude) but can even directly affect intention or behavior.

Second, this study found, for the first time, that the relationship between perceived policy effectiveness and attitude is negatively and significantly moderated by waste management knowledge, which implies that the effect of perceived policy effectiveness on attitude would decrease with an increase in waste management knowledge. The result confirms that human behavior in the ABC theory is the result of a complex interaction between internal and external factors [41]. This can be explained as the fact that in Shanghai, most residents have a wealth of waste management knowledge, and they are already familiar with how to sort waste. In this case, the government repeatedly emphasizing the benefits of waste sorting to residents with high waste-management knowledge will be of little significance and may even arouse the residents’ resentment because it takes up their time [66]. As Jackson [67] believes, policies that encourage individuals to restrain certain behaviors can only achieve limited success. Therefore, when knowledge can effectively promote residents to form a positive attitude, the impact of perceived policy effectiveness will be weakened. However, it is worth noting that the perceived policy effectiveness measured by this study was limited to the government’s financial investment and policy propaganda effect. This does not mean that the relationship between the promotion of other types of policies and residents’ attitudes will be weakened by an increase in waste management knowledge. Furthermore, this study also confirmed that knowledge had a significant and positive moderating effect on the link between implementation intention and proenvironmental behavior. This shows that knowledge and implementation intention are indispensable for proenvironmental behavior. Proenvironmental behavior tends to appear in people with high knowledge and strong implementation intentions.

### 6.2. Practical Implications

Considering the positive impact of perceived policy effectiveness on attitude, implementation intention, and proenvironmental behavior, the government should continue to increase policy support to promote the formation of a new pattern of environmental protection and sustainable development. To improve the effectiveness of the policy, the government should not only increase financial investment and strengthen policy publicity in the short term but, more importantly, improve collection and transportation procedures and improve terminal disposal facilities in the long term [68]. Source separation affects terminal disposal, and the terminal disposal situation, in turn, affects residents’ enthusiasm for source separation. If source separation and terminal disposal work well together, a virtuous cycle will be formed; otherwise, it will be a vicious cycle. The high cost and low efficiency of waste disposal facilities are important factors hindering the waste management process. Therefore, the policy should cover the entire process of waste management, including source separation, collection, transportation, and terminal disposal, and not just the supervision and restriction of residents’ behavior.

The positive influence of knowledge on the relationship between implementation intention and proenvironmental behaviors has also been proven in this study. Although Shanghai residents generally know how to sort waste, this does not mean that publicity and education on waste sorting can be stopped. As Asia’s leading country in waste management, Japan has introduced environmental education at the kindergarten stage and has continuously optimized its environmental education system [69]. Developed countries use systematic environmental education to help people understand environmental knowledge and form environmental protection awareness from an early age, develop environmental protection habits, and drive changes in society through changes from generation to generation. The Shanghai government should still require schools, at all stages, to provide environmental protection courses and include them in the assessment system. Moreover, considering the population mobility of Shanghai, for new residents who have just moved to Shanghai, it is recommended that the community where they live becomes responsible for the publicity and guidance on waste sorting. For residents who do not cooperate, the community has the right to hold them accountable and request corrections.

The Shanghai government should take the existing waste sorting problems in Shanghai seriously. For example, the classification of recyclable waste is not as simple as that of dry and wet waste. Although recyclable waste will not pollute other types of waste (e.g., wet waste), when they are mixed with other waste, the utilization rate of recyclables is greatly reduced, which is not conducive to resource recycling and sustainable development. The government should increase subsidies for different types of recyclables, including low-value recyclables, for example, setting up small recycling sites in various communities, where residents can exchange the collected recyclables for daily necessities or coupons. Moreover, the penalties for residents who do not sort recyclables should be the same as those for residents who do not sort wet waste.

## 7. Conclusions

In response to the proposed research objectives, this study points out that under the influence of mandatory policies, more than 90% of Shanghai residents often or always sort dry and wet waste, which is much higher than that of residents in other cities (e.g., Changsha, Xiamen). At present, Shanghai’s waste sorting policy is quite effective, and such a good momentum should be maintained and Shanghai’s experience should be extended to other cities. Moreover, the results demonstrate that perceived policy effectiveness, knowledge, and attitude are important factors that affect residents’ implementation intention and proenvironmental behavior. In particular, perceived policy effectiveness can affect proenvironmental behavior directly or through the mediating effects of attitude and implementation intention. Therefore, this study recommends that the Shanghai Municipal Government continue to strengthen the breadth and depth of policy support and policy publicity. The government should not only pay attention to source separation but also improve collection, transportation, and terminal disposal.

This study has certain limitations. First, more than 80% of the respondents in this study had a college/university degree or above; hence, the respondents’ education level was relatively high, which may have a certain impact on the survey results. Future research should pay more attention to low-income, low-educated groups and compare the behaviors of different groups. Second, the survey on policy in this study only considered a relatively shallow level. Future research should consider more aspects of policy support, including satisfaction with rewards/punishments, facilities, and staff. Third, this study does not provide alternative models, and future research can provide more possibilities based on this.

## Figures and Tables

**Figure 1 ijerph-18-02910-f001:**
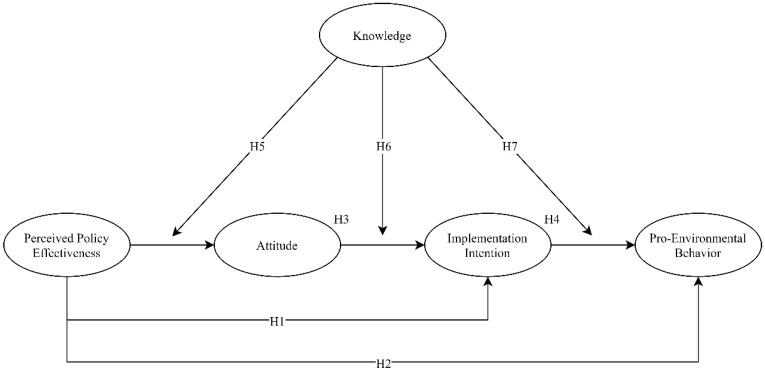
The hypothesized model.

**Figure 2 ijerph-18-02910-f002:**
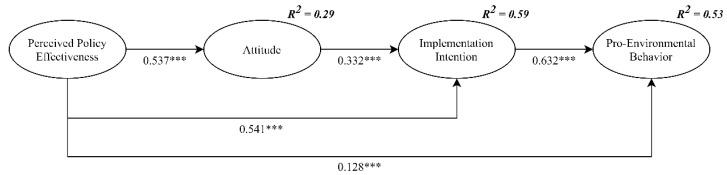
Structural path model. *** *p* < 0.001. Standardized coefficients are reported.

**Figure 3 ijerph-18-02910-f003:**
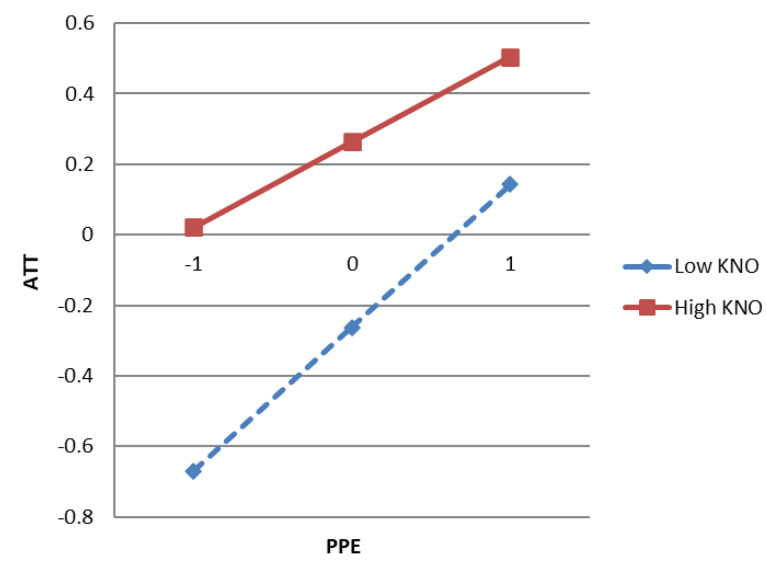
The interactive effect of knowledge on the relationship between perceived policy effectiveness and attitude.

**Figure 4 ijerph-18-02910-f004:**
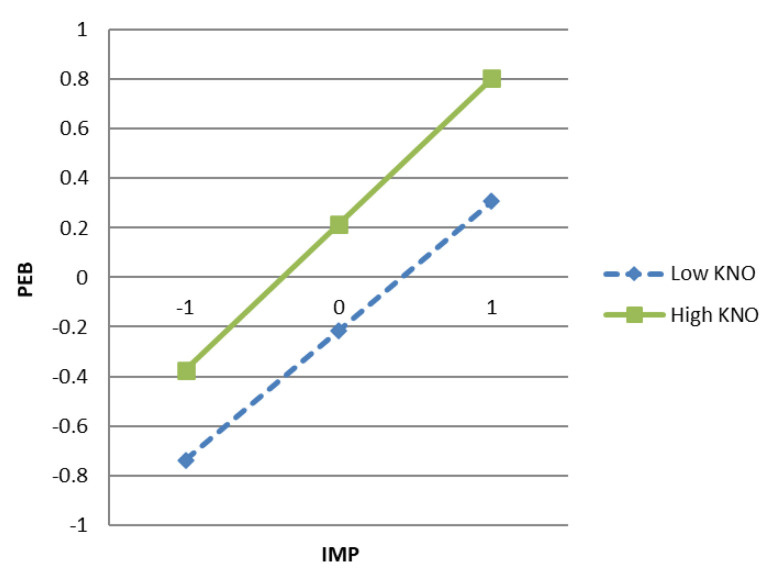
The interactive effect of knowledge on the relationship between implementation intention and proenvironmental behavior.

**Table 1 ijerph-18-02910-t001:** Questionnaire’s response rates of Shanghai residents.

District Zone	District	Population (Million) ^a^	Distributed	Returned
Eastern SH	Pudong	5.550	900	478
Southern SH	Fengxian	1.152	200	114
Western SH	Qingpu	1.219	200	136
Northern SH	Baoshan	2.042	300	145
Central SH	Minhang	2.544	400	272
	Total	12.507 ^a^	2000	1145

SH = Shanghai. ^a^ Shanghai Municipal Statistics Bureau [50].

**Table 2 ijerph-18-02910-t002:** Participant profile (N = 1145).

Profiles	Survey	Census ^a^
Respondent age (%)		
≤17	10.1	≤14 (10.1%)
18–28	27.2	15–64 (73.8%)
29–44	33.0	-
45–59	19.0	-
≥60	10.7	≥65 (16.1%)
Respondent gender (%)		
Male	52.6	49.5
Female	47.4	50.5
Respondent education level (%)		
Below high school	9.4	
High school/Vocational school	10.4	
College/University	62.2	
Master or Ph.D.	18.0	
Monthly salary (%)		
≤5000 CNY	35.2	Mean 8765 CNY
5001–10,000 CNY	29.4	
10,001–20,000 CNY	23.0	
≥20,001 CNY	12.3	

^a^ Shanghai Municipal Statistics Bureau [50].

**Table 3 ijerph-18-02910-t003:** Reliability and validity tests.

Items	Loadings	Cα	AVE	CR
Perceived policy effectiveness		0.920	0.641	0.925
PPE1: The Government has increased financial investment to support waste sorting.	0.687			
PPE2: The environmental programs organized by the Government have effectively aroused environmental awareness in the general public.	0.802			
PPE3: The Government provides clear guidelines and regulations on waste sorting.	0.856			
PPE4: The Government campaign helps citizens understand the importance of waste sorting.	0.886			
PPE5: The Government campaign clearly explains the benefits of waste sorting.	0.878			
PPE6: The Government promotes waste sorting as a positive symbol, label, image, and event.	0.869			
PPE7: The Government’s policy facilitates me in the separation and recycling of household waste.	0.574			
Attitude		0.905	0.762	0.906
ATT1: Waste sorting is sensible.	0.863			
ATT2: Waste sorting is useful.	0.861			
ATT3: My feelings towards waste sorting are favorable.	0.894			
Implementation intention		0.889	0.729	0.890
IMP1: For the next garbage discard, I plan to separate everything in advance when I am at home.	0.876			
IMP2: For the next garbage discard, I plan to put paper waste and plastic bottles into the recycling bin provided by the Government.	0.827			
IMP3: For the next garbage discard, I plan to put wet waste into the designated trash can within the stipulated time.	0.858			
Proenvironmental behavior		0.891	0.628	0.894
PEB1: During the previous week, how often did you separate wet waste?	0.753			
PEB2: During the previous week, how often did you separate dry waste?	0.777			
PEB3: During the previous week, how often did you recycle paper and paper products?	0.795			
PEB4: During the previous week, how often did you separate recyclable bottles (e.g., plastic bottles, aluminum/tin cans, glass bottles) and containers?	0.815			
PEB5: During the previous week, how often did you separate waste for recycling purposes?	0.821			

All standardized loadings are significant at the 0.001 level.

**Table 4 ijerph-18-02910-t004:** Proenvironmental behavior and frequency.

Variables	Categories (%)				
	Never	Rarely	Sometimes	Often	Always
How often do you separate wet waste?	0.4%	1.4%	7.2%	40.0%	51.0%
How often do you separate dry waste?	0.5%	0.7%	5.5%	40.4%	52.8%
How often do you recycle paper and paper products?	0.8%	3.2%	13.3%	36.3%	46.4%
How often do you separate recyclable bottles and containers?	0.7%	2.6%	12.2%	38.0%	46.5%
How often do you separate waste for recycling purposes?	1.5%	2.6%	13.6%	36.9%	45.4%

**Table 5 ijerph-18-02910-t005:** Discriminant validity test.

Construct	Mean	SD	PPE	ATT	IMP	PEB
PPE	4.122	0.657	**(0.801)**			
ATT	4.391	0.754	0.511 **	**(0.873)**		
IMP	4.326	0.653	0.665 **	0.558 **	**(0.854)**	
PEB	4.315	0.666	0.533 **	0.416 **	0.659 **	**(0.792)**

The square root of the average variance extracted (AVE) is in diagonals (bold); off diagonals are Pearson’s correlations of constructs. ** *p* < 0.01.

**Table 6 ijerph-18-02910-t006:** Standardized direct, indirect, and total effects.

	Point Estimate	Product of Coefficients		Bootstrapping				
				Percentile 95% CI		Bias-Corrected 95% CI		Two-Tailed Significance
		*SE*	*Z*	Lower	Upper	Lower	Upper	
Direct effects								
PPE → IMP	0.541	0.040	13.525	0.456	0.613	0.461	0.619	0.000 (***)
PPE → PEB	0.128	0.044	2.909	0.040	0.215	0.039	0.215	0.004 (**)
Indirect effects								
PPE → IMP	0.178	0.031	5.742	0.124	0.247	0.125	0.248	0.000 (***)
PPE → PEB	0.454	0.039	11.641	0.378	0.531	0.384	0.537	0.000 (***)
Total effects								
PPE → IMP	0.719	0.026	27.654	0.664	0.765	0.666	0.767	0.000 (***)
PPE → PEB	0.583	0.031	18.806	0.520	0.640	0.520	0.640	0.000 (***)

Standardized estimation of 5000 bootstrap samples; ** *p* < 0.01, *** *p* < 0.001.

**Table 7 ijerph-18-02910-t007:** Hierarchical regression analysis.

Variable	ATT		IMP		PEB	
	Step 1	Step 2	Step 1	Step 2	Step 1	Step 2
Constant	0.000	0.047				
PPE	0.352 ***	0.324 ***				
KNO	0.276 ***	0.263 ***				
PPE × KNO		−0.082 ***				
Constant			0.000	−0.001		
ATT			0.369 ***	0.370 ***		
KNO			0.495 ***	0.395 ***		
ATT × KNO				0.001		
Constant					0.000	−0.019
IMP					0.536 ***	0.556 ***
KNO					0.215 ***	0.215 ***
IMP × KNO						0.033 *
F	259.117	185.378	433.678	288.870	498.164	335.066
R^2^	0.312	0.328	0.432	0.432	0.466	0.468
ΔR^2^	0.312	0.016	0.432	0.000	0.466	0.002

** p* < 0.05, *** *p* < 0.001. Unstandardized coefficients are reported.

## Data Availability

Not applicable.

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
