# Peer review of "Linking Perceived Policy Effectiveness and Proenvironmental Behavior: The Influence of Attitude, Implementation Intention, and Knowledge"

_ijerph, 2021, doi:10.3390/ijerph18062910_

Round 1

Reviewer 1 Report

The study examined the relationship between perception of policy, attitude, knowledge, implementation intentions, and pro-environmental behavior in a group of Shanghai residents. The authors found attitude mediated the direct relationship, and knowledge moderated the relationship between implementation intention and PEB). Overall, it is a timely topic, and it applies to the local ecology. The authors collected a large residential sample. However, I have a few concerns about the conceptual and statistical approaches.

The authors used TPB as an overall model in the study and then explained each studied variable/hypothesis independently. How does each variable (e.g., perception of policy effectiveness, attitude, knowledge, implementation intention, and pro-environmental behavior) fit the TPB? How does the moderated mediation model fit the TPB?

The authors theorized that attitude was a mediator and knowledge was a moderator. However, from the introduction (section 3.2 and 3.3), both variables could be mediators (to explain the link between perception of policy and behavior) or moderators (to increase the strength of the relationship). How was the specific path determined? Did the authors examine alternative models? For an example of one alternative model, why not examine the relationship between attitude and PEB with policy perception as a moderator or a mediator?

On a similar note, only two articles were cited under each section (3.2 and 3.3). Many studies have conducted on the topic of TPB in environmental psychology/education using attitude and knowledge. These studies were not reviewed.

Two surveys were sent out to participants. Did the authors use the first survey for main variables and the second survey for actual behaviors? Did all participants return both surveys? What was the attribution rate between stage one and stage two?

About 10% of the survey was not from adults (age < = 17 years old). Did they also respond to the question about income and education? They had less income or no income, and they did not have a college degree. Also, did the authors request parental consent for this age group?

My understanding is not all items were used in the original measures (e.g., line 262). If it is the case, please specifically discuss each measure why and how certain items were excluded/included in the current study.

I didn’t get the measurement model in section 5.2. Convergent validity typically means if different measures (e.g., PEB, pro-environmental intentions, pro-environmental attitudes…) are similar and correlated. It could be used within the measures (different items). But researchers typically examine item analysis (e.g., Differential Item Functioning Test) among different items in a measure across different groups. I am not sure factor loadings can do such things. Did the authors obtain the factor loadings from EFA or CFA? What measurement model did the authors run?

Moderator was tested in regression with interaction terms. Why not running the moderated mediation analysis in the Process macro in SPSS or R. There are many advantages to running it using a regression (e.g., what about multicollinearity issue since the correlations were very high?).

There were only two citations in the entire discussion section. The authors should go back to the theory (e.g., TPB) and discuss how the project (direct path, mediator, and moderator) supports the theory and the past research.

Incorrect citations. E.g., line 269 should be cited as #39; line 266 should be #40. Please check the references and in-text citations throughout the manuscript.

On line 263, table 3 should be table 4?

Figures are missing from the manuscript.   

Reviewer 2 Report

Very interesting research and data analyses.

Line 178: pro-implementation intention or implementation intention?Line 180: environmental behaviour or pro-environmental behaviour?

Please explain in more details on data collected during Stage 1 and Stage 2 . It is a question if it is a longitudinal research as different data were collected during stage 1 and stage 2. How many participants were in Stage 1, Stage 2, and both. What was an attrition rate from stage1 to stage 2.

How about parents' consent for their children participated in survey (10 % <18)?

Round 2

Reviewer 1 Report

Nice revision. Please add the model number on line 338